# A new design of wind power prediction method based on multi-interaction optimization informer model

Wenjuan Zhou[1,2]*, Feng Huang[1,2], Bing Wei[1,2], Liang Li[1,2], Shixi Dai[1,2], Xin Xie[1,2], Youyuan Peng[1,2], Hong You[1,2]

**1** Hunan Institute of Engineering, School of Electrical and Information Engineering, 88 Fuxing East Road, Yuetang, Xiangtan, Hunan, China, **2** Chongqing Jiangdong Machinery Co, Ltd., Chongqing, China

* 2208085810@qq.com

## Abstract

The accurate prediction of wind power is imperative for maintaining grid stability. In order to address the limitations of traditional neural network algorithms, the Informer model is employed for wind power prediction, delivering higher accuracy. However, due to insufficient exploration of dynamic coupling among multi-source features and inadequate data health status perception, both prediction accuracy and computational efficiency deteriorate under complex working conditions.This study proposes a prediction framework for the Informer model based on multi-source feature interaction optimization (MFIO-Informer). Integrating physical feature collaborative analysis with data health status perception has been shown to enhance prediction accuracy and reduce computation time. First, the Lasso algorithm and Pearson correlation coefficient method are applied to screen key multi-source features from wind turbine operation and maintenance data, quantifying their dynamic correlations with power output. Secondly, a fully-connected neural network (FNN) is employed to establish a hidden coupling model of wind speed, blade deflection angle, and power for extracting the Dynamic Synergistic Coefficient (DSC), which characterizes equipment performance. Subsequently, a health assessment of wind turbine data is conducted, leveraging historical power data and DSC. This assessment yields a health matrix, which is instrumental in optimizing the encoding, decoding, and embedding vector prediction processes of the Informer model. Finally, power prediction experiments are conducted on two public wind power datasets using the proposed MFIO-Informer model.The experimental results demonstrate that, in comparison with the traditional Informer model, the MFIO-Informer model attains approximately 20% higher prediction accuracy and 54.85% faster prediction speed.

**Data availability statement:** The two datasets used in the paper, including: Data set 1. Data set of KDD CUP 2022 Dataset. Data set 2. Data set of "China Software Cup" - Longyuan Wind Power Race Track Dataset. have all obtained permission and consent from the relevant authors. Under the premise of non-profit research, these two datasets can be made publicly available. All dataset files are available from the Data Review URL: https://aistudio.baidu.com/datasetdetail/352838

**Funding:** The author(s) received no specific funding for this work.;

**Competing interests:** The authors have declared that no competing interests exist.

## Introduction

Fluctuations in wind cause instability in wind power generation [1], affecting grid operation after connection. To ensure grid safety, power prediction and timely regulation are essential [2]. However, the strong randomness of wind power (which is affected by environmental factors such as wind speed and turbulence) and the progressive degradation of equipment performance (e.g., blade wear and gearbox aging) present dual challenges for traditional prediction models operating under complex conditions: decreased accuracy and inadequate timeliness.

In recent years, deep learning models (e.g., long short-term memory (LSTM) and transformer) have significantly improved the accuracy of wind power prediction by capturing temporal dependencies [3]. Nevertheless, traditional models are limited in modeling multi-source feature coupling and data health perception, which restricts their practical application [4]. To enhance prediction speed and accuracy, researchers have explored two major approaches: First, statistical learning methods (e.g., ARIMA and support vector machines) simplify prediction through linear assumptions, yet they struggle to characterize nonstationary correlations between wind speed and power [5]. Second, deep neural networks (e.g., CNN-LSTM hybrid models) excel at mining local spatiotemporal features, yet they are limited in their ability to perceive data health [6].

Deep neural networks use fixed encoding structures that lack the ability to adapt dynamically to wind power equipment degradation. This compromises prediction accuracy. Recently, the Transformer variant Informer model reduced computational complexity with ProbSparse self-attention and demonstrated advantages in long-sequence prediction. This model shows great potential in wind power forecasting [7]. However, the Informer model has several limitations: First, input features are mostly concatenated trivially, failing to model the physical synergies among multi-source data (e.g., wind speed and blade angle). Second, the attention mechanism ignores the evolution of wind turbine health status, leading to error accumulation in long-term predictions [8]. Third, the static parameter allocation of the encoder-decoder makes balancing computational efficiency and prediction accuracy difficult. These issues render existing methods inadequate for meeting the high demands of real-time scheduling and intelligent operation and maintenance in wind farms.

To improve prediction performance, researchers optimized Informer in several ways. Liu, F., Wang, X., Gong, M., et al. [9–11] improved the encoding method to reduce prediction error rates in long-term, multidimensional, strongly correlated sequences. They also strengthened the encoder module's encoding capability and enhanced Informer's prediction accuracy. Long, H., Cao, Y., Yang, Z., et al. [12–14] introduced an adversarial mechanism to strengthen the decoder module's ability to resist interference, mitigate the impact of placeholders and autoregressive effects, prevent error accumulation, and improve prediction accuracy. Wu, Bradateanu, and Li [15–17] improved the sampling strategy to focus on strongly correlated features, reduce the computational complexity of self-attention, accelerate sparse matrix sampling, and increase Informer's prediction speed. Currently, most optimization methods

fail to enhance both Informer's prediction accuracy and speed simultaneously. These methods produce unsatisfactory optimization results, increase prediction costs, and hinder Informer's practical application.

This article is structured into five sections: The first section presents the overall methodological framework. The second section describes the principles of model construction. The third section details the comprehensive optimization scheme and its validation. The fourth section provides case verifications and result analyses. The fifth section offers the conclusion.

## 1. Methodological framework

The principle of the algorithm is illustrated in Fig 1, comprising the following steps:

(1) Integrate the original dataset to obtain multi-source data,including humidity, yaw angle, blade deflection angle, wind speed, and temperature, from the target wind turbine.

(2) Preprocess the data by removing null values and outliers and normalizing it to meet the requirements of the neural network input.

(3) Apply the Lasso algorithm and the Pearson correlation method to select features from wind turbine O&M data and eliminate irrelevant features.

(4) Train an FNN using the filtered data to derive the functional relationship between power output and operational features.

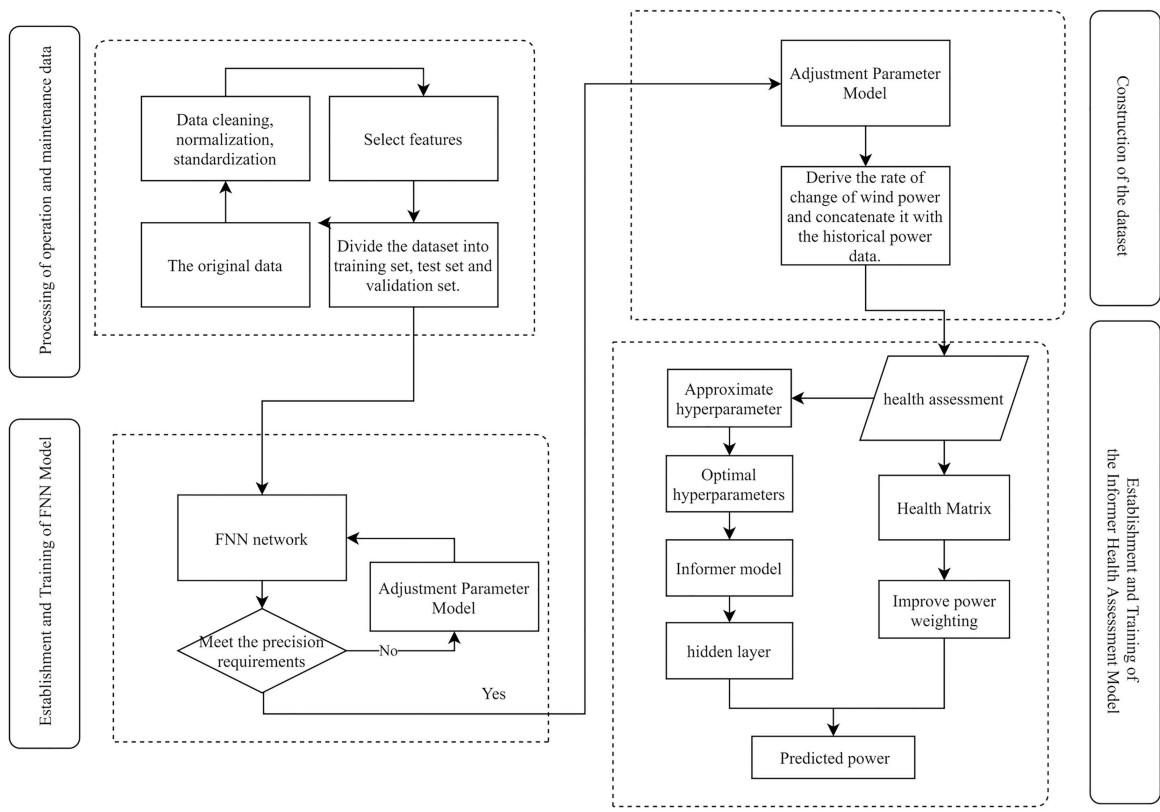

**Fig 1. Flowchart of algorithm principles.**

 

(5) Calculate the dynamic synergy coefficient $k$ of wind power using the trained FNN and the chain rule.

(6) Conduct a health assessment by combining $k$ with historical power values.

(7) Construct a health matrix, adjust weight allocation, optimize hyperparameters, and train the Informer network.

(8) Optimize the predicted power output, generate test results for a comparative analysis, and draw conclusions.

The principle of the power prediction method is shown in Fig 1.

## 2. Principles of model construction

### 2.1. Feature selection via lasso and pearson correlation methods

The Lasso algorithm performs feature selection and model sparsification through L1 regularization. The core principle is to impose an L1 norm constraint on the regression model, which forces some feature coefficients to zero and eliminates redundant features. The penalty function refines the model by compressing or setting to zero the coefficients of less relevant features when multiple features exist. The remaining non-zero coefficients correspond to highly correlated features, thereby achieving feature selection. To mitigate the impact of feature scale disparities on regression performance, the algorithm standardizes the predictors so that they have a mean of 0 and a variance of 1.

The Pearson correlation method measures vector correlation, with outputs ranging from −1 to +1: A value of 0 indicates no correlation, negative values denote negative correlation, and positive values signify positive correlation. [18]The formula is as follows:

$$Pearson = \frac{\sum (x_i - \bar{x}) \sum (y_i - \bar{y})}{\sqrt[2]{\sum (x_i - \bar{x})^2} \sqrt[2]{\sum (y_i - \bar{y})^2}}$$

(1)

In the formula, the closer the value of $i$ is to 1, the higher the correlation.

Using a combination of the Pearson correlation and Lasso algorithms to analyze wind power data filters out irrelevant features and reduces dimensionality. The results are shown in Figs 2–3.Table 1 shows the specific values of the stronger correlations found among wind speed $(S)$, blade deflection angle $(\theta)$, and power $(P)$.

### 2.2 Construction and training of FNN-based wind power prediction model

With [St, θt] as the input space and [Pt] as the output space, the training, validation, and test sets are divided in the ratio 8:1:1. A 2-500-50-1 feed-forward neural network (FNN) architecture is constructed, adopting the adaptive moment estimation (Adam) optimizer and mean squared error (MSE) loss function for training. An early-stopping strategy is implemented, whereby iteration halts when the loss difference between the training and test sets is less than 5%. The FNN adjusts the model weights using the backpropagation (BP) algorithm.

The FNN model was evaluated as follows: the adjusted determination coefficient $R^2$ on the test set was 0.9921, and the root mean square error ($RMSE$) was 0.0288.

### 2.3. Construction and training of FNN prediction model

The data objects—wind speed $(S)$, blade deflection angle $(\theta)$, and power $(P)$—are continuous and cannot undergo abrupt changes. Because continuity implies differentiability, the partial derivative of power with respect to wind speed can be derived using a feed-forward neural network (FNN):

$$P' = \frac{\partial F(S,\theta)}{\partial s}$$

(2)

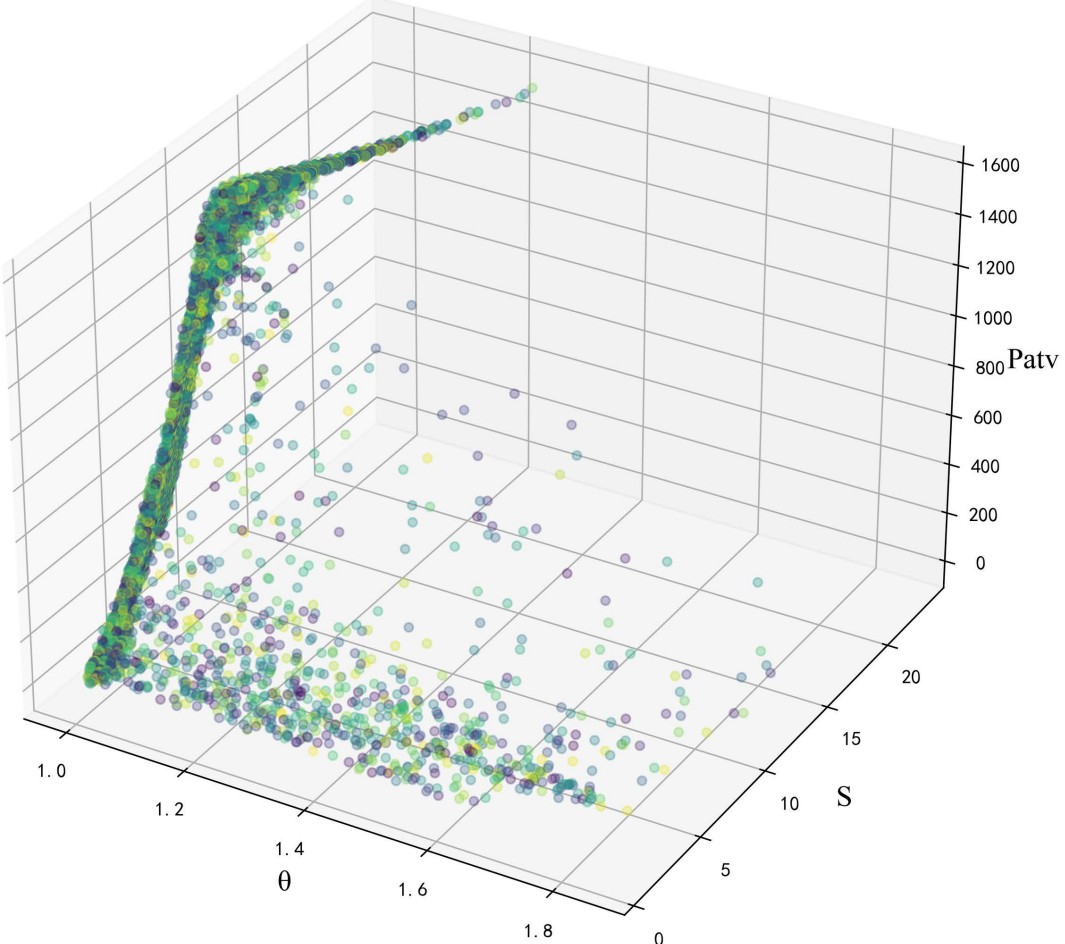

**Fig 2. Three-dimensional scatter plot of wind speed, blade deflection angle, and power.**

In the formula, $S$ denotes wind speed.

Direct differentiation of implicit functions is unfeasible, but when applying the BP algorithm, the loss function ($L$) performs global partial derivation, enabling calculation of $\partial P/\partial L$ and $\partial S/\partial L$. Assuming $L$ is the mean squared error ($MSE$) loss function, and since $P_i$ is a constant, the chain rule further yields $P'$ as:

$$P' = \frac{\partial F(S,\theta)}{\partial S} = \frac{1}{n} \cdot \frac{S_{grad}}{2F(S,\theta)}$$

(3)

In the formula, $n$ denotes the data volume, but in this formula, $n$ is set to 1.

Calculate the rate of change of the power generation at this moment with respect to wind speed, and concatenate the power and the rate of change at the corresponding moment into vectors $[P'_t, P_t]$.

Taking the data of the No. 1 wind turbine as an example, the calculation results $P'_t$ are shown in Table 2.

The derivative rate represents the partial derivative of active power with respect to wind speed ($\partial P/\partial S$), calculated based on the proposed FNN model. From this, the dynamic coordination coefficient and the changing trend of power $P$ under a certain $S$ can be obtained.

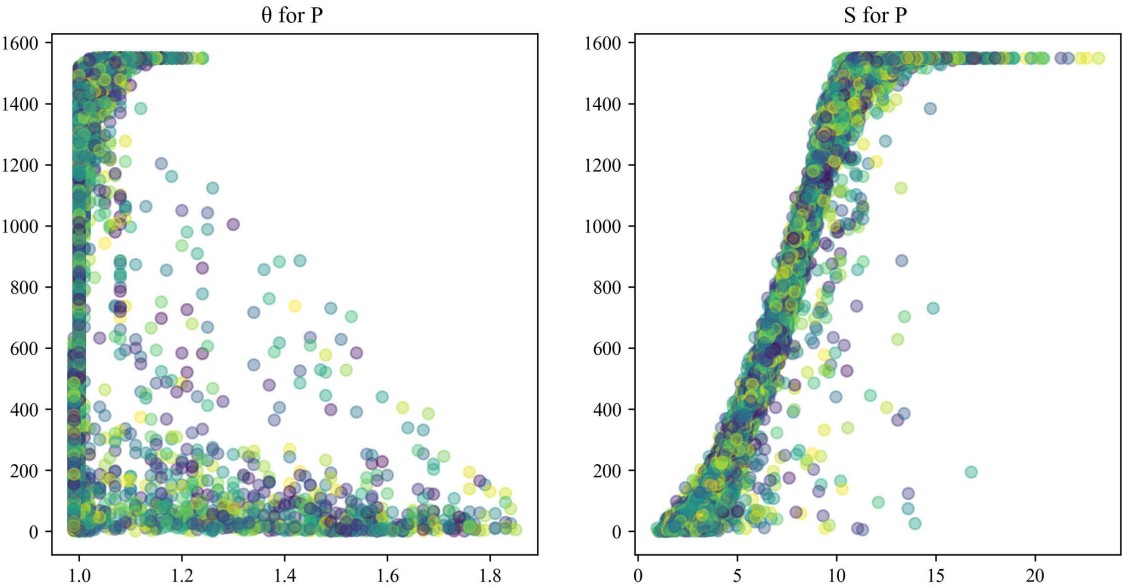

**Fig 3. Two-dimensional scatter plot of wind speed, blade deflection angle, and power.**

**Table 1. Comparison of correlation.**

| Feature Name | Lasso Coefficients | Pearson Correlation |
| --- | --- | --- |
| Wind Speed (Wspd) | 147.772 | 0.965 |
| Blade Deflection Angle (Pad) | −534.477 | 0.033 |
| Ambient Temperature (Etmp) | −20.740 | −0.027 |
| Internal Turbine Temperature (Itmp) | 21.604 | 0.153 |
| Wind Direction (Wdir) | 0.714 | −0.069 |
| Yaw Angle (Ndir) | 0.006 | −0.182 |

**Table 2. Sample data of wind speed, blade angle, active power, and derivative rate.**

| Wind speed S (m/s) | Blade angle θ(·) | Active power Patv (kW) | Derivative Rate (kW/(m/s)) |
| --- | --- | --- | --- |
| 11.64 | 12.95 | 703.74 | 2.96 |
| 10.95 | 11.38 | 837.15 | 3.12 |
| 10.48 | 9.91 | 875.01 | 3.46 |
| 10.95 | 10.16 | 993.20 | 3.69 |

## 2.4. Informer power prediction model

Informer is a self-attention-based prediction model proposed by Zhou Haoyi et al. in 2021 [18]. Unlike conventional prediction models, it features a separated encoder-decoder structure. During prediction, the encoder and decoder only interact via coding output transmission, with no feedback mechanisms involved in any prediction steps.

In the Informer, there are three key structures:

1. **Self-attention Distilling Operation**: The encoder module reduces the length of the output sequence layer by layer to extract strongly correlated features. After feature connection, the encoded outputs are concatenated as inputs to the decoder module.[19]

2. **Generative Style Decoder**: Using masked multi-head attention for decoding, this structure enables simultaneous prediction of multiple tokens. Unlike traditional sequential token prediction (where each step's output is fed back, causing loss accumulation in long sequences), it mitigates error propagation and preserves predictive accuracy..

3. **ProbSparse Self-Attention Mechanism:** By focusing on high-score dot products, Informer prioritizes strongly correlated features during self-attention. Sampling only these features reduces computational complexity and prediction cost.

The principle of the Informer model is illustrated in Fig 4.
The existing Informer neural network model exhibits three fundamental limitations:

(1) **Frequent Attention Overhead**: The Encoder module necessitates secondary dot-product calculations to accentuate strongly correlated features, resulting in a substantial reduction in prediction speed.[20,21].

(2) **Decoding Imbalance**: The Decoder utilizes sequence placeholders for auxiliary prediction; however, equal weighting of placeholders and feature data results in accuracy degradation due to unreasonable weight distribution.[22,23]

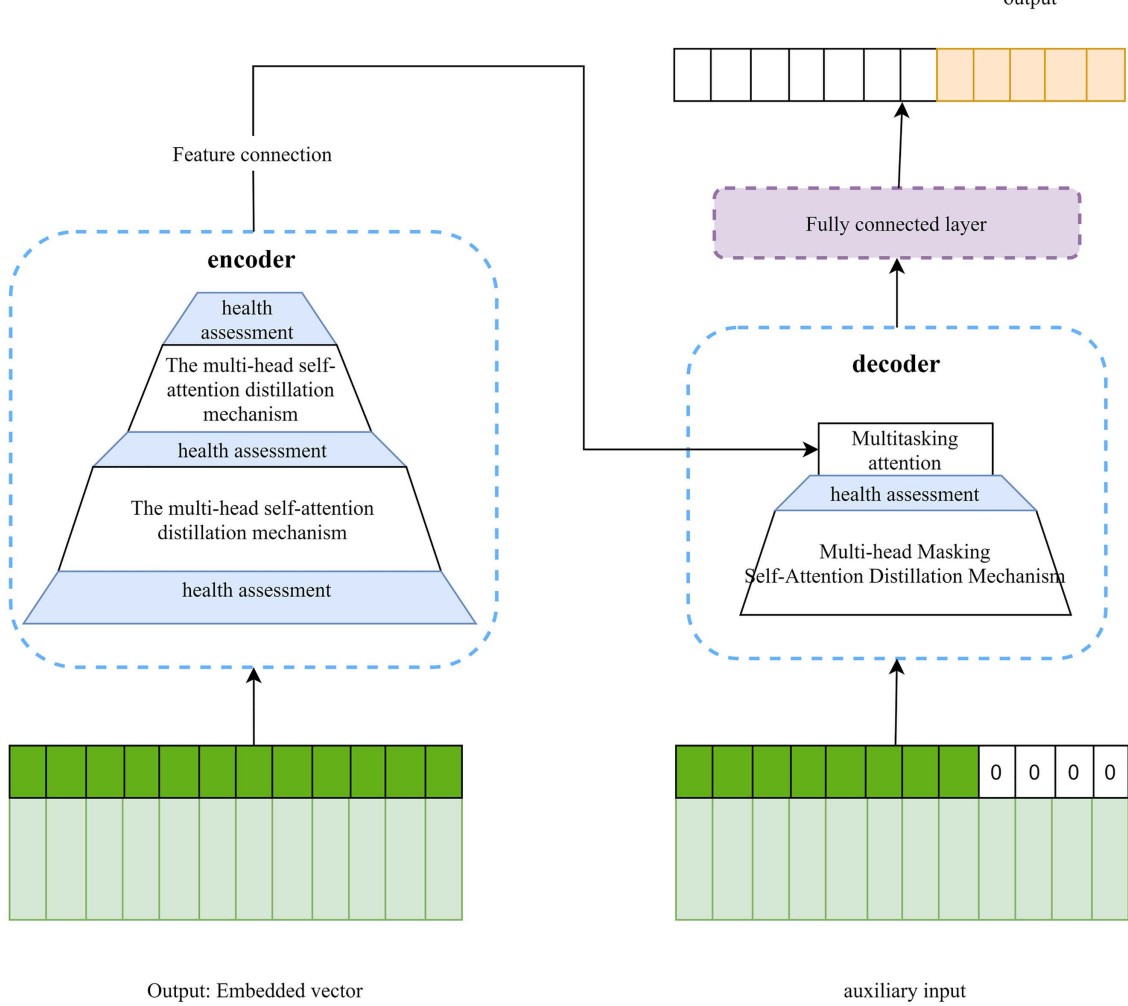

**Fig 4. Flowchart of informer prediction.**

(3) **Computational Complexity**: Implementing the self-attention mechanism necessitates additional modules or feature dimension expansion, increasing computational complexity and compromising the model's lightweight design [24,25].

In order to address these constraints and enhance prediction accuracy and speed, the Informer model is proposed to be optimized through multi-source data reconstruction and wind turbine health assessment. By leveraging a health matrix to refine the encoder, decoder, and embedding vector processes, the optimized MFIO-Informer model is developed to balance computational efficiency and predictive performance.

### 2.5. Principles of health matrix evaluation

Health assessment, a weight – optimization algorithm proposed by Lars Landberg in 2011, transforms a two – dimensional probability density function into a weight – based health matrix. This transformation is carried out according to the statistical probability distribution characteristics of the model output.

(1) **Data Collection**: Collect operating data of wind turbines, such as wind speed, temperature, pressure, rotational speed, and power, through sensors and SCADA systems.;

(2) **Data Cleaning**: Clean missing values, abnormal values, and transmission noise in the operating data to enhance data reliability.;

(3) **Feature Selection:** From the cleaned operating data, select feature data. Then, determine the dimension of the health matrix based on the dimension of the selected feature data;

(4) **State Classification:** Use the feature data to calculate feature weights. Classify the states of wind turbines (e.g., normal, minor fault, severe fault) according to the magnitude of these weight values [26];

The formula for calculating feature weights is as follows:

$$f(x; \mu, \Sigma) = \frac{1}{(2\pi)^{\frac{i}{2}} |\Sigma|^{\frac{1}{2}}} exp \left[ \frac{1}{2} (x - \mu)^T \Sigma^{-1} (x - \mu) \right]$$

(4)

In the formula, $i$ represents the feature index, where $i$ ranges from 1 to $n$, $\mu$ is the mean vector, and $\Sigma$ is the covariance matrix.

(5) **Matrix Generation**: The feature weights calculated via Equation (4) are arranged sequentially to form a weight matrix $Q_{n \times 1}$. Each element in $Q$ represents the weight coefficient of the corresponding feature in the same state. Subsequently, factor analysis is applied to the operational data to generate a correlation matrix $R_{1 \times n}$. Finally, the health matrix $A$ is obtained by multiplying $Q$ and $R$, expressed as:

$$A_{n \times n} = Q_{n \times 1} R_{1 \times n}$$

(5)

In the formula, $n$ denotes the dimension of the health matrix, typically set to 6–12.

The health assessment workflow is illustrated in Fig 5.

The optimization analysis of health assessment is as follows:

Let the first-layer input of the prediction model be $z^{(0)}$, with a coefficient matrix $D^0_{n \times n}$ where the principal diagonal elements are 1 and the remaining elements are within (0, 1). The row and column labels of $D^0$ are sorted to maintain consistent ordering.

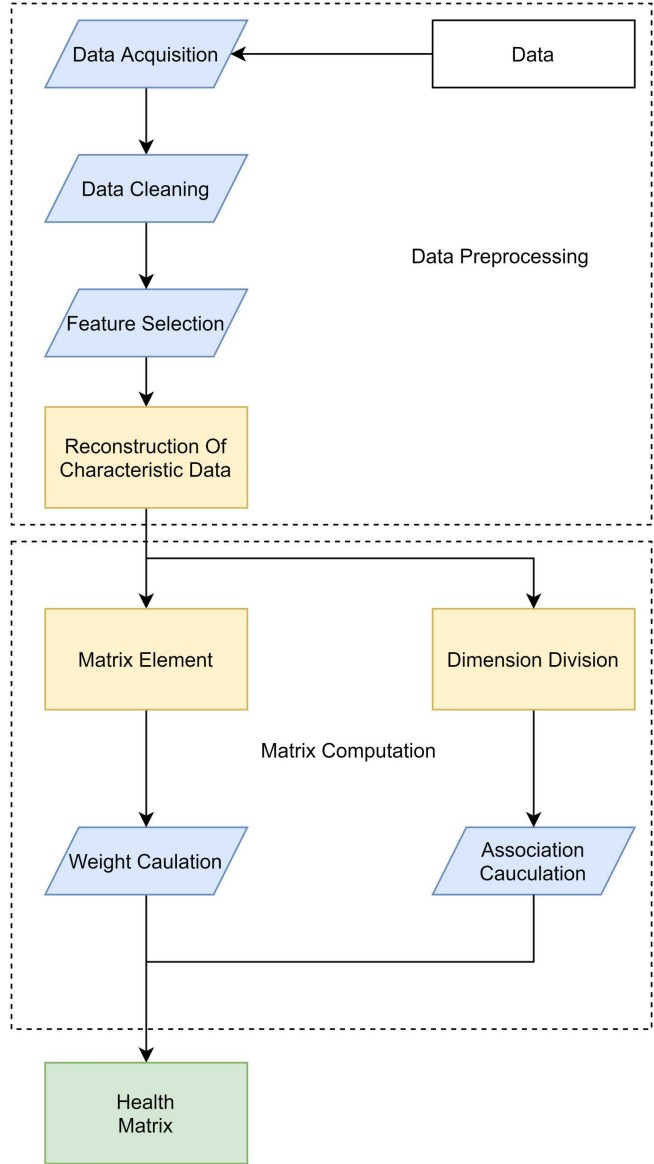

**Fig 5. Health assessment process diagram.**

During prediction, the coefficient matrix $D^i$ for the $i$-th layer input $z^{(i-1)}$ can be viewed as the $i$-th power of $D^0$:

$$D^i = [D^0]^i = D^0 \times D^0 \times \cdots \times D^0 \qquad (6)$$

The health assessment optimization is realized by multiplying the input $z^{(0)}$ with the health matrix $A$ to obtain $z_1^{(0)}$, i.e., $z_1^{(0)} = z^{(0)} \times A$. The coefficient matrix of $z_1^{(0)}$, denoted as $D_1^0$, is calculated by:

$$D_1^0 = D^0 \times A \qquad (7)$$

Here, $A$ reduces the weights of weakly correlated features in $D^0$. While maintaining diagonal dominance and symmetry, $A$ adjusts the corresponding principal elements of $D^0$ to be less than 1. By using $D_1^0$ instead of $D^0$, each forward computation decreases the weights of weakly correlated features once, thereby mitigating their influence and enhancing the model's prediction performance.

## 3 Optimization plan

### 3.1 Encoding optimization

The Encoder module consists of multiple encoding layers and convolutional layers. The input of the i-th layer is $x^{(i-1)}$ and the output is $x^{(i)}$. In each encoding layer, $x^{(i-1)}$ undergoes forward propagation through Multi-Head Self-Attention, followed by dimensionality reduction via a convolutional layer to obtain $x^{(i)}$, expressed as:

$$x^{(i)} = Conv(Attention(x^{(i-1)}))$$

(8)

Here, Conv denotes convolutional computation in the convolutional layer, and Attention represents the ProbSparse self-attention function:

$$p = softmax\left(\frac{\overline{Q}K^T}{\sqrt{d_k}}\right)V$$

(9)

In Equation (9), softmax is the normalization function, $\overline{Q}$ is the probability sparse matrix sampled from $x^{(i-1)}$, $\sqrt{d_k}$ is the scaling factor for key-value sampling, and $K$, $K^T$, $V$ are key-value adjustment matrices.

All features in $x^{(i-1)}$ initially have equal weights. To emphasize strongly correlated features, the input is weighted as:

$$x_1^{(i-1)} = x^{(i-1)} \times A$$

(10)

In Equation (8), replacing $x_1^{(i-1)}$ with $x^{(i-1)} = x^{(i-1)} \times A$ enhances the weights of strongly correlated features, accelerating the sampling and generation of $\overline{Q}$, and improving the Encoder's encoding speed.

Encoding optimization analysis:

In Equation (9), the *i*-th query element of $\overline{Q}$ isderived via:

$$p(q_i, K, V) = \sum_j \frac{k(q_i, k_j)}{\sum_l k(q_i, k_l)} v_j = Exp_{p(k_j|q_i)}[v_j]$$

(11)

Where $q_i, k_i, v_i$ denote the *i*-th row of $\overline{Q}$ $K$, $V$,respectively.The asymmetric exponential kernel $k(q_i, k_j) = Exp(q_i k_j^T / \sqrt{d})$, and sparse metric $M(q_i, K)$ for *Top-u* query,are defined as:

$$M(q_i, K) = \max_j \left\{\frac{q_i k_j^T}{\sqrt{d}}\right\} - \frac{1}{L_K}\sum_{j=1}^{L_K} \frac{q_i k_j^T}{\sqrt{d}}$$

(12)

In the formula,$L_K$ is the sequence length of matrix $K$.

From Equations (8) and (9), it is known that the in Equation (10) increases the weights of strongly correlated features and will be input as the input of each layer of the Encoder module. Thus, the queries involved $M(q_i, K)$ in Equations (8), (9), (11), and (12) only need to perform a single dot product calculation, and the computational complexity of sparse self-attention is reduced to $O(\frac{L}{2} \ln L)$, thereby improving the encoding speed and the prediction speed of Informer.

In sparse self-attention, the lengths of queries and keys are both equal to $L$, leading to a computational complexity of $O(L \ln L)$. However, when all feature weights in $x^{(i-1)}$ are equal, the $M(q_i, K)$ query requires a secondary dot-product calculation to enhance strongly correlated features, which decreases the Encoder's encoding speed.

From Equations (8) and (9), weighting $x_1^{(i-1)}$ with $A$ (as in Equation (10)) emphasizes strongly correlated features. Using $x_1^{(i-1)} = x^{(i-1)} \times A$ as the Encoder's input allows the $M(q_i, K)$ queries in Equations (8), (9), (11), and (12) to require only single dot-product calculations. This reduces the computational complexity of sparse self-attention to $O(\frac{L}{2} \ln L)$, thereby enhancing the Encoder's encoding speed and Informer's prediction efficiency.

## 3.2. Decoding optimization

Let the Decoder module comprise $L$ decoding layers. In each layer, the input $c_{L-1}$ and feature data $y$ are processed through multiple decoding sub-layers, expressed as:

$$c_L = Decoder(y, c_{L-1}) = Decoder_L(Decoder_{L-1}(\cdots Decoder_1(y, c_0)\cdots)) \tag{13}$$

In, the weights of all features are equal. In the consideration of improve $c_{L-1}$, the weights of strongly correlated features, that is,

In the formula, $Decoder_i(\cdot)$ denotes the decoding calculation of the $i$-th layer, $c_0$ is the input of the first layer, $c_{i-1}$ is the input of the $i$-th layer, and $y$ represents the feature data at the prediction time.

All features in $c_{L-1}$ initially have equal weights. To enhance the weights of strongly correlated features in $c_{L-1}$, we define:

$$c_L = Decoder(y, c_{L-1}) = Decoder_L(Decoder_{L-1}(\cdots(Decoder_1(y, c_0) \cdot h)\cdots)) \tag{14}$$

In the formula, $h$ denotes the health factor, derived from the compressive transformation of the health matrix $A_{n \times n}$. The output x of each layer, when multiplied by the health factor $h$, serves as the input for the subsequent layer.

The input to the first layer of the Decoder module $c_0$ can also be expressed as:

$$X_{de}^t = Concat(X_{token}^t, X_0^t) \in R^{(L_{token}+L_y) \times d_{model}} \tag{15}$$

In the formula, $X_{token}^t$ denotes the encoded outputs of the Encoder module, and $X_0^t$ is a sequence placeholder with a scalar value of 0 (corresponding to the all-zero part of the auxiliary input). Here, $L_{token}$ represents the sequence length of the encoded output, $L_y$ denotes the sequence length of the placeholder, $d_{model}$ is the feature dimension, and *Concat* indicates vector concatenation. By setting the placeholder and masking the dot product to $-\infty$, Informer applies masked multi-head attention in the Decoder's decoding process.

Although the masked dot product prevents autoregression (where each feature focuses on its next timestamp in $X_{de}^t$), the equal weighting of $X_{token}^t$ and $X_0^t$ causes cumulative errors from $X_0^t$ to propagate during each decoded sublayer's forward computation. This error diffusion degrades the Decoder's decoding accuracy.

The optimization in Equation (13) effectively increases the weight of $X_{token}^t$ while reducing that of $X_0^t$. As shown in Equations (8) and (9), each forward decoding iteration in Equation (14) mitigates cumulative errors, thereby enhancing the Decoder's decoding accuracy. The improved decoding accuracy directly boosts Informer's prediction accuracy after optimization.

## 3.3. Embedding vector optimization

Embedded vectors serve as the input to the first layer of the Encoder module. As the Encoder module progressively reduces the length of the encoded sequence layer-by-layer to extract strongly correlated features, we enhance the weights of these features through optimization, as shown in Equation (16):

$$x_1^{(0)} = x^{(0)} \times A \tag{16}$$

In the formula, $x^{(0)}$ is the unoptimized embedding vector, and $x_1^{(0)}$ represents the optimized embedding vector. After optimization, $x_1^{(0)}$ replaces $x^{(0)}$ as the input to the first layer of the Encoder module.

Let $X_t$ be the *t-th* input sequence, where the global timestamp type of $X_t$ is $p$, and the feature dimension being $d_{model}$. The embedding vector that preserves local context is obtained through fixed-position embedding using the following two formulas:

$$PE_{(pos,2j)} = \sin(pos/(2L_x)^{2j/d_{model}}) \tag{17}$$

$$PE_{(pos,2j+1)} = \cos(pos/(2L_x)^{2j/d_{model}}) \tag{18}$$

In the formulas, *POS* represents the learnable embedding using global timestamps, with a limited vector length (up to 60, in minute units). To align dimensions, the Informer model projects the scalar context into a $d_{model}$-dimensional vector via a 1D convolution filter (kernel width = 3, stride = 1).

Another representation of the embedding vector $x^{(0)}$ is as follows:

$$x_{feed[i]}^t = \alpha u_i^t + PE_{(L_x \times (t-1)+i)} + \sum_p \left[SE_{(L_x \times (t-1)+i)}\right]_p \tag{19}$$

In the formula, $i \in \{1, \cdots, L_x\}$ are the amplitude factors between the balanced scalar projection and the local/global embedding. If the sequence has been normalized, $\alpha = 1$.

The structure of the embedding vector is illustrated in Fig 6.

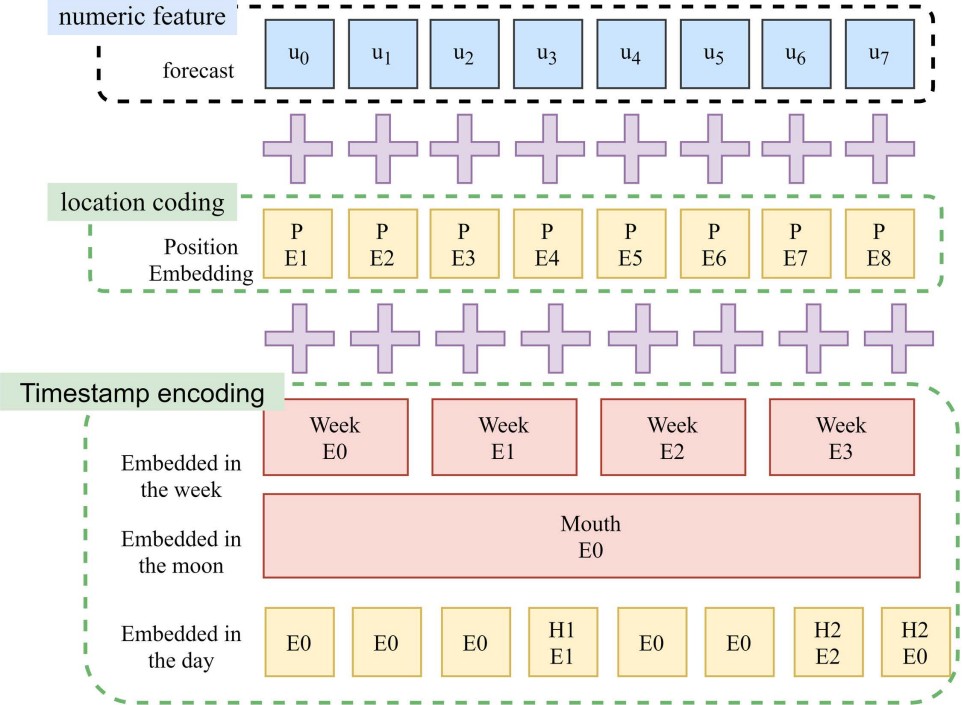

**Fig 6. Structure diagram of embedded vectors.**

It is evident that the inherent sampling weights of trigonometric sine and cosine functions, as demonstrated in Equations (14) and (15), result in a divergent weight distribution from the original data. As illustrated in Fig 3, the integration of local and global timestamps cannot achieve a one-to-one correspondence, thereby introducing systematic errors. As demonstrated in Equations (8), (9), (10), (11), and (12), these systematic errors undergo a gradual expansion during the ensuing encoding-decoding process, thereby impacting the prediction accuracy of Informer.

Equation (13) optimizes the embedding vector by accounting for feature distribution changes. According to Equations (8) and (9), systematic errors arising from feature distribution changes gradually diminish in the subsequent encoding-decoding process after optimization, thereby improving Informer's prediction accuracy.

## 4. Instance verification

### 4.1. Error metrics

Three error indicators are employed: $R_2$ (Coefficient of Determination), *MSE* (Mean Squared Error), and *MAE* (Mean Absolute Error), as defined in Equations (20)–(22):

$$R^2 = 1 - \frac{\sum_{i=1}^{n} (\hat{y}_i - y_i)^2}{\sum_{i=1}^{n} (\overline{y}_i - y_i)^2}$$

(20)

$$MSE = \frac{1}{n} \sum_{i=1}^{n} (\hat{y}_i - y_i)^2$$

(21)

$$MAE = \frac{1}{n} \sum_{i=1}^{n} |\overline{y}_i - y_i|$$

(22)

The formulas define $y_i$ as the true value, $\hat{y}_i$ as the predicted value, and $\overline{y}_i$ as the true mean. $R^2$ (Coefficient of Determination) measures goodness-of-fit, where values closer to 1 indicate better model consistency with observed data. MSE (Mean Squared Error) and MAE (Mean Absolute Error) quantify prediction dispersion—smaller values signify lower prediction fluctuations.

For effective optimization, ideal scenarios show increasing $R^2$ alongside decreasing MSE and MAE. However, due to varying prediction dispersions across models [24], enhancing overall fit might compromise local fitting at specific points, potentially increasing outlier handling costs [27]. Optimization is still considered effective if:

- Two metrics improve by ≥10% relative to the baseline, and

- The third metric decreases by ≤2%.

### 4.2. Simulation testing

The experimental phase entailed the execution of tests employing two publicly accessible datasets: the Baidu KDD CUP 2022 dataset and the "China Software Cup"—Longyuan Wind Power Track dataset. Each dataset contained between 17,000 and 21,000 pieces of data.

To facilitate a comprehensive comparison of the optimization effects, eight power prediction models were selected for simulation testing and comparison: no DSC optimization, with DSC optimization, no DSC-Encoder optimization, with DSC-Encoder optimization, no DSC-Encoder-Decoder optimization, with DSC-Encoder-Decoder optimization, no DSC-embed optimization, and with DSC-embed optimization. Each model was subjected to three rounds of training and

six rounds of training. The initial 80% of the dataset was allocated for training, while the remaining 20% was designated for evaluation. The test platform comprised the following components: 12th Gen Intel(R) Core(TM) i5-12500H 2.50 GHz CPU, 16 GB RAM, 200 GB HDD, and the operating system was Windows 11.

In order to mitigate the impact of units, dimensions, and other variables, the dataset, prediction data, and related materials were standardized according to the following protocol:

$$z_1 = \frac{1}{z_s}(z_0 - \bar{z})$$

(23)

In the formula, $z_0$ represents the unstandardized prediction result, $z_1$ represents the standardized prediction result, $z_s$ represents the variance of the prediction result, and $\bar{z}$ represents the average value of all prediction results.

The component details of the Informer are shown in Table 3.

The prediction results based on the KDD CUP dataset are shown in Figs 7–10:

The prediction results based on the Longyuan Wind Power dataset are shown in Figs 11–14.

In Figs 7–14, V1 in Tables 4–7 denotes the prediction performance without DSC and any optimization, V2 represents the performance with DSC but without optimization, V3 signifies the performance without DSC and with no Encoder optimization, V4 indicates the performance with DSC but without Encoder optimization, V5 shows the performance without DSC but with Encoder-Decoder combination optimization, V6 represents the performance with both DSC and Encoder-Decoder combination optimization, V7 denotes the performance without DSC but with embed optimization, and V8 signifies the performance with both DSC and embed optimization.

As observed from Figs 7–14, the prediction accuracy of all optimized models except the Encoder-optimized one has improved compared to the unoptimized models. Tables 4–7 reveals that the seven proposed optimized models have achieved at least one of the following optimization effects: a 20% improvement in prediction accuracy or a 54.85% improvement in prediction speed compared to the unoptimized models. Furthermore, the three Encoder-Decoder combination optimized models have simultaneously achieved a 15% improvement in prediction accuracy and a 50%

**Table 3. Informer component details.**

| | Encoder module |
|---|---|
| **Input** | 1×3 convolutional layer (stride 1, kernel width 3) |
| | Embedding vector (dimension d = 512) |
| **Probabilistic sparse self-attention block** | Multi-head sparse self-attention (number of heads h = 16, dimension d = 32), dropout rate p = 0.1<br>Probabilistic sparse network (dimension d = 2048), with the activation function being the GELU function, and the dropout rate being p = 0.1 |
| | A 1×3 convolutional layer (with stride 1 and kernel width 3), using the ELU activation function<br>Max pooling (stride = 2) |
| **Distillation** | Decoder module<br>1×3 convolution layer (stride = 1, kernel width = 3) |
| | Initial input (dimension d = 512) |
| | Adding mask on the attention block |
| **Input** | Multi-head self-attention (number of heads h = 8, dimension d = 64), dropout rate p = 0.1 |
| | Probabilistic sparse network (dimension d = 2048), with the activation function being the GELU function, and the dropout rate being p = 0.1 |
| **Masking** | Encoder module |
| **Self-attention block** | 1×3 convolutional layer (stride 1, kernel width 3)<br>Embedding vector (dimension d = 512) |
| | Multi-head sparse self-attention (number of heads h = 16, dimension d = 32), dropout rate p = 0.1 |

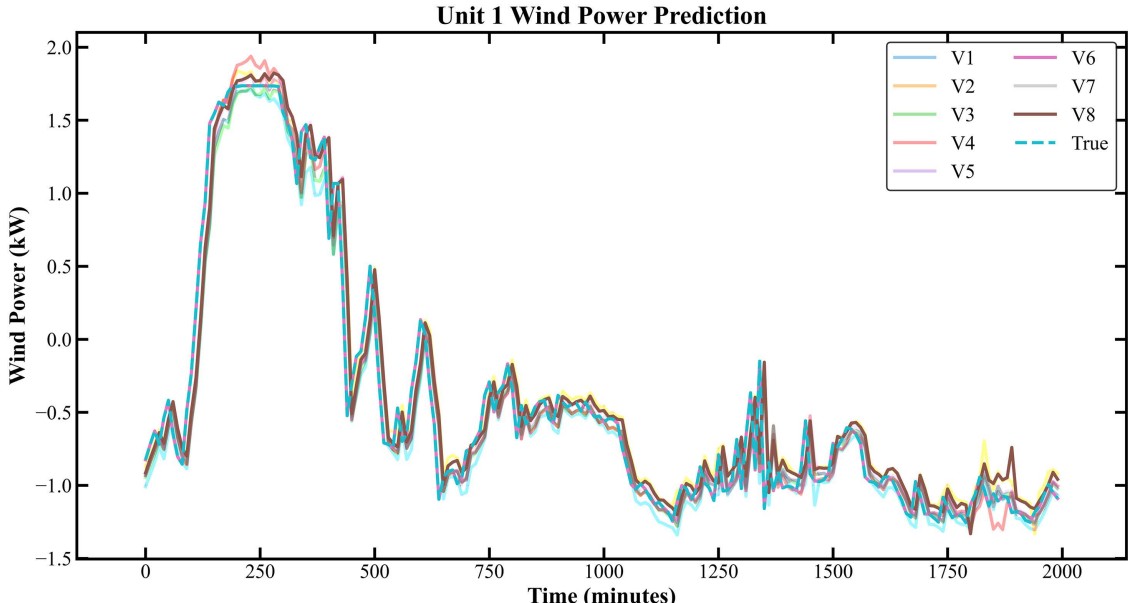

**Fig 7. Prediction Results of Three Rounds of Training for Unit K-C 1.**

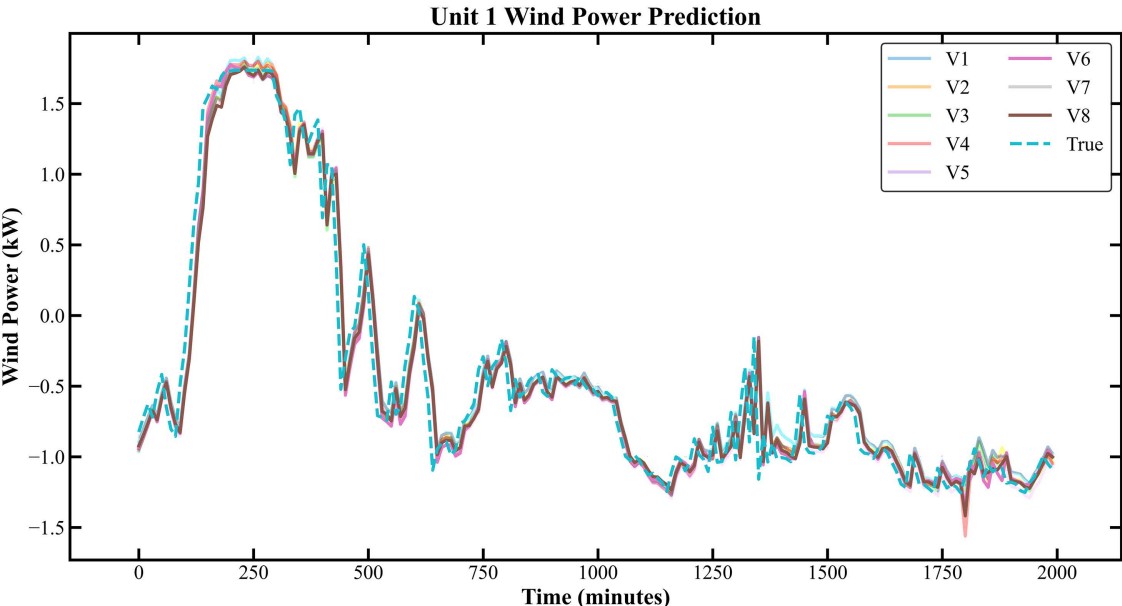

**Fig 8. Prediction Results of 6 Rounds of Training for Unit K-C 1.**

improvement in prediction speed. Additionally, the selected combined optimized models require only half the number of training rounds, and their prediction accuracy has already approached or exceeded that of the unoptimized models. This demonstrates that the Informer model based on multi-variable interaction optimization is reasonable and effective, and the MFIO-Informer model is a high-quality wind power prediction model.

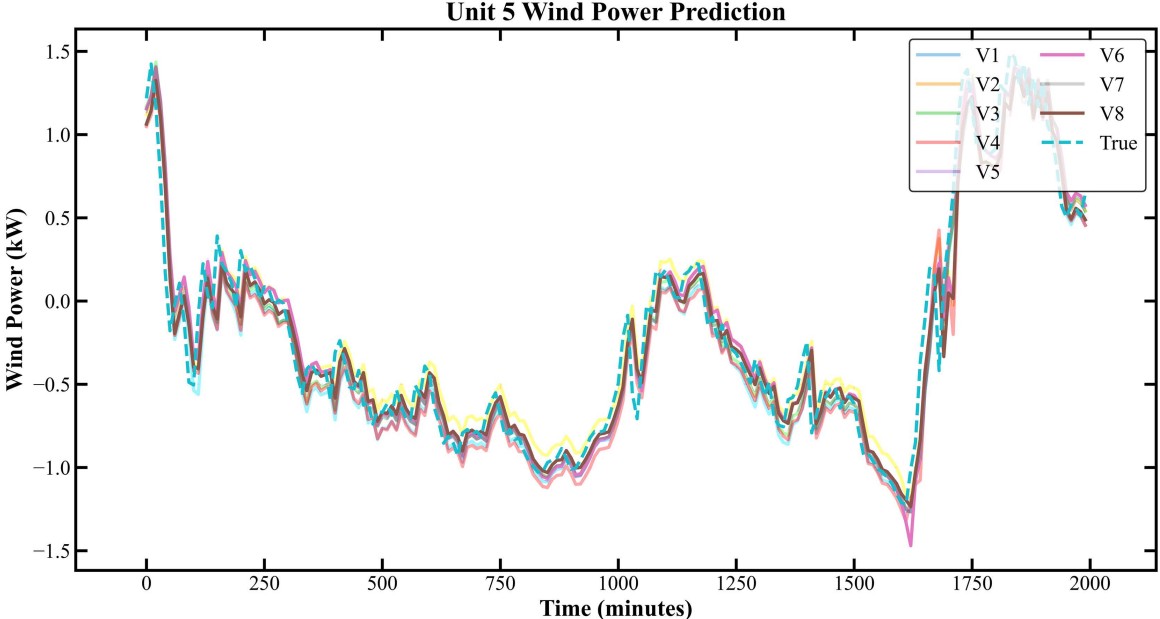

**Fig 9. Prediction Results of Three Rounds of Training for Unit 5 of K-C Power Plant.**

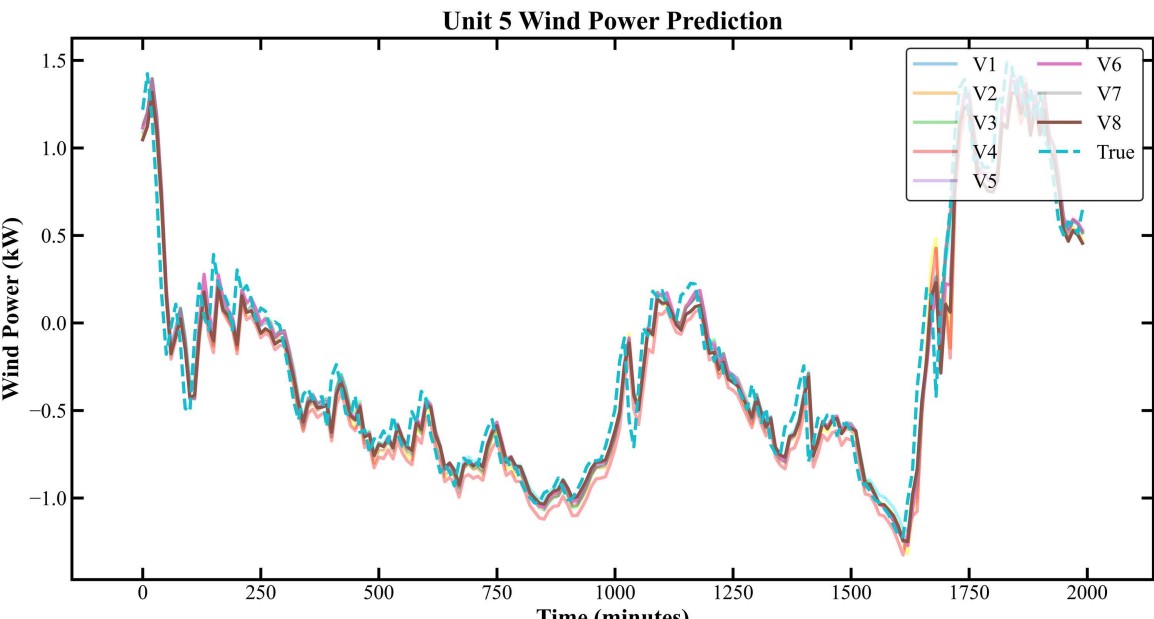

**Fig 10. Prediction Results of 6 Rounds of Training for Unit 5 of K-C Power Plant.**

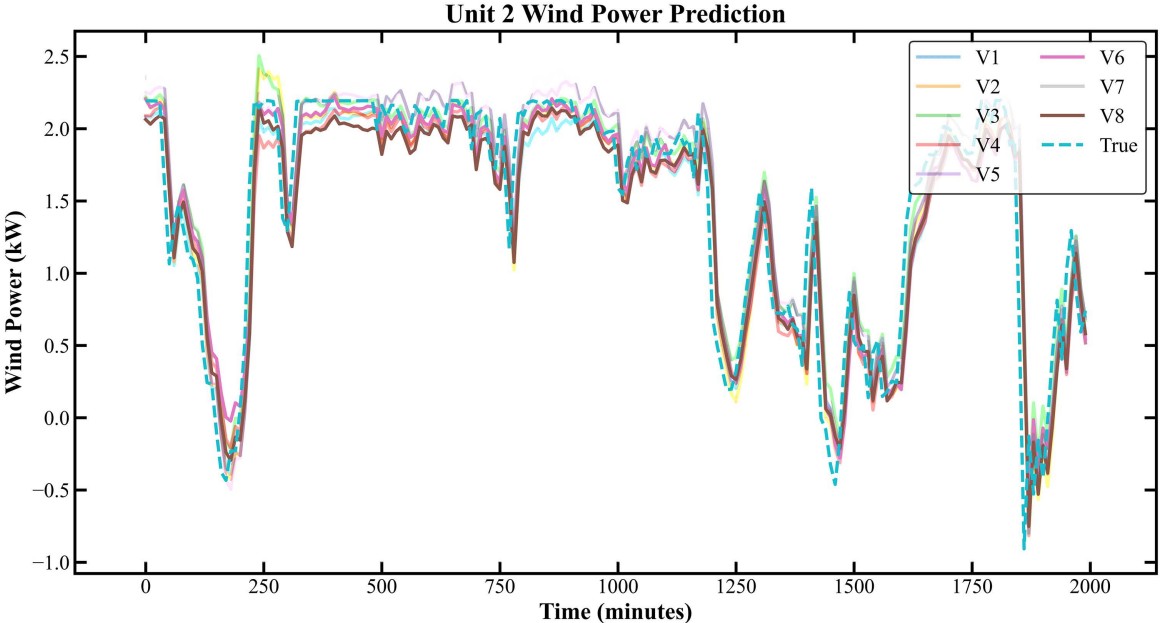

**Fig 11. Prediction Results of Three Rounds of Training for Unit 2 of Longyuan Power Station.**

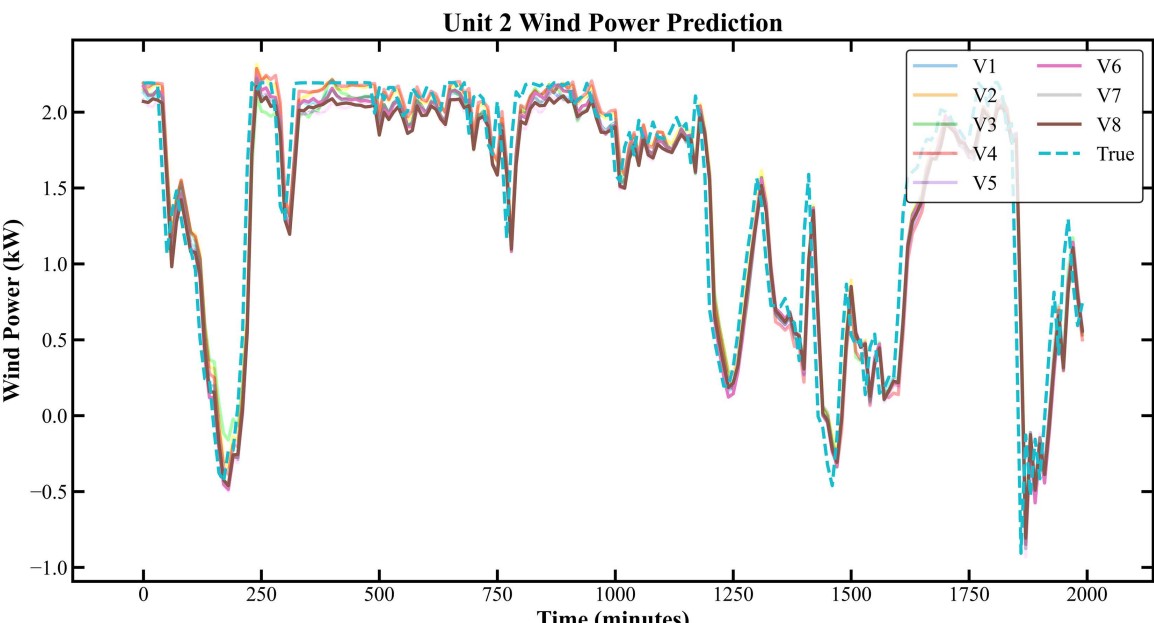

**Fig 12. Prediction Results of 6 Rounds of Training for Unit 2 of Longyuan Power Plant.**

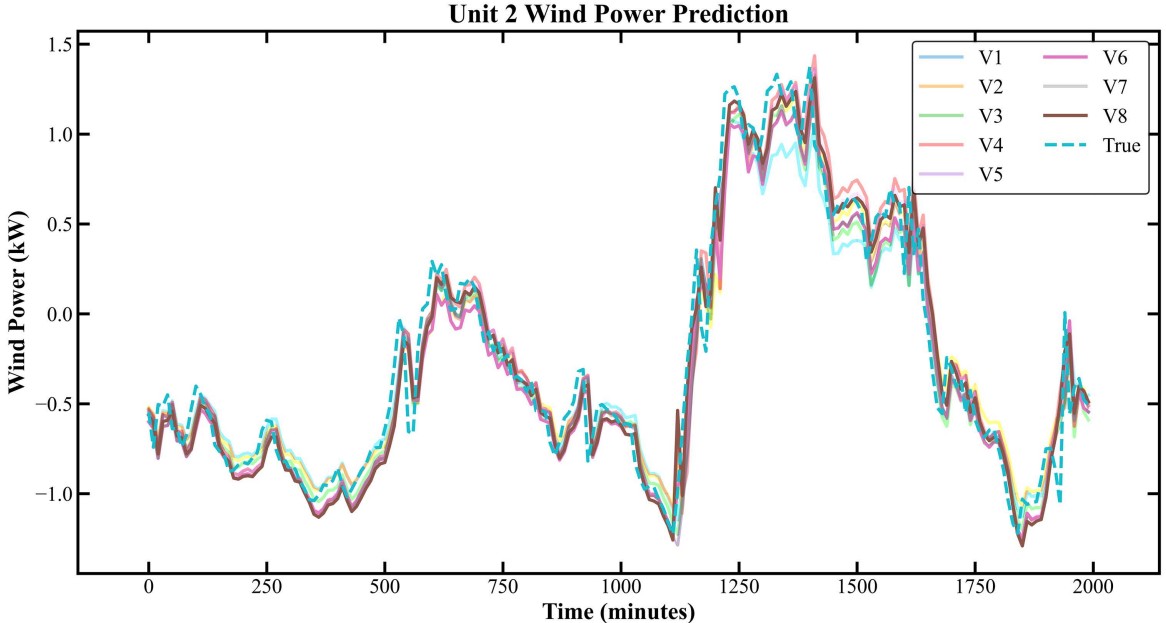

**Fig 13. Prediction Results of Three Rounds of Training for Unit 13 of Longyuan Power Station.**

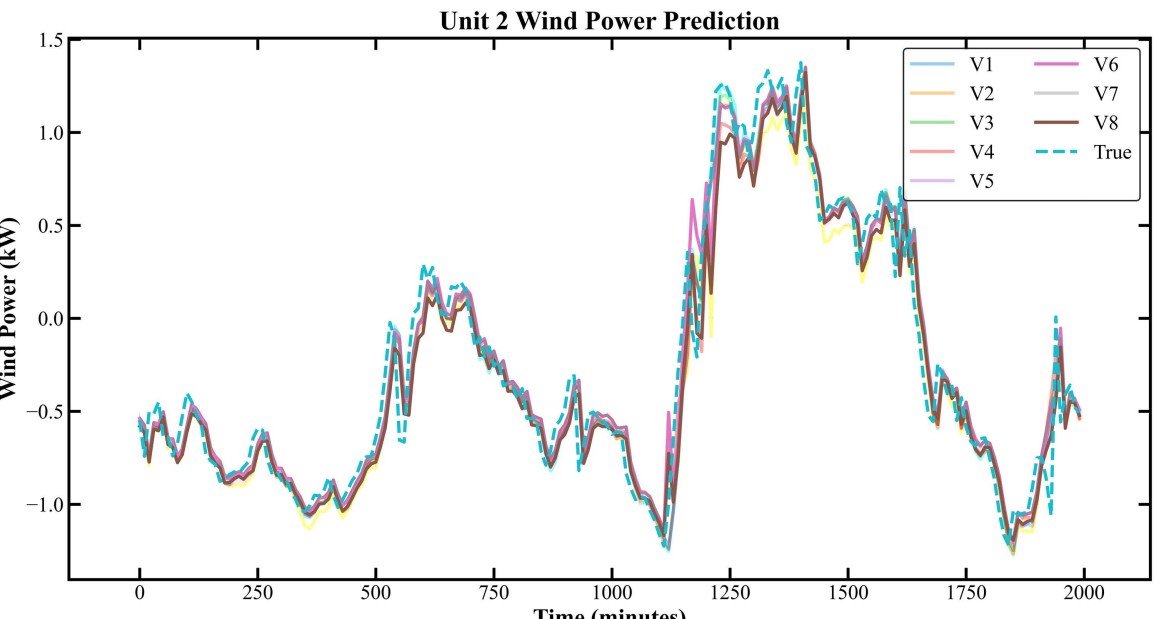

**Fig 14. Prediction Results of 6 Rounds of Training for Unit 13 of Longyuan Power Plant.** The prediction indicators of each dataset are presented in Tables 4–7.

**Table 4.** Prediction indicators of unit 1 of KDD CUP.

**3 Rounds of Training**

|  | R2 | MSE | MAE | Time/s |
|---|---|---|---|---|
| V1 | 0.9198 | 0.0690 | 0.1795 | 126.7032 |
| V2 | 0.9276 | 0.0619 | 0.1577 | 101.4570 |
| V3 | 0.9236 | 0.0646 | 0.1628 | 68.2020 |
| V4 | 0.9297 | 0.0648 | 0.1607 | 65.8393 |
| V5 | 0.9224 | 0.0710 | 0.1636 | 67.8995 |
| V6 | 0.9374 | 0.0664 | 0.1602 | 67.7608 |
| V7 | 0.9278 | 0.0632 | 0.1576 | 69.6268 |
| V8 | 0.9321 | 0.0622 | 0.1530 | 66.2020 |

**6 Rounds of Training**

|  | R2 | MSE | MAE | Time/s |
|---|---|---|---|---|
| V1 | 0.9296 | 0.0624 | 0.1559 | 122.9593 |
| V2 | 0.9316 | 0.0621 | 0.1542 | 142.6094 |
| V3 | 0.9295 | 0.0621 | 0.1546 | 67.1701 |
| V4 | 0.9324 | 0.0620 | 0.1537 | 67.3861 |
| V5 | 0.9309 | 0.0616 | 0.1539 | 91.5230 |
| V6 | 0.9432 | 0.0614 | 0.1531 | 90.1255 |
| V7 | 0.9267 | 0.0615 | 0.1582 | 100.9162 |
| V8 | 0.9485 | 0.0613 | 0.1546 | 102.4420 |

**Table 5.** Prediction indicators of unit 5 in KDD CUP.

**3 Rounds of Training**

|  | R2 | MSE | MAE | Time/s |
|---|---|---|---|---|
| V1 | 0.9257 | 0.0754 | 0.1807 | 134.1894 |
| V2 | 0.9259 | 0.0710 | 0.1771 | 133.6501 |
| V3 | 0.9305 | 0.0697 | 0.1701 | 67.8995 |
| V4 | 0.9240 | 0.0742 | 0.1846 | 68.0760 |
| V5 | 0.9138 | 0.0681 | 0.1690 | 95.6929 |
| V6 | 0.9498 | 0.0710 | 0.1710 | 97.1340 |
| V7 | 0.9331 | 0.0694 | 0.1703 | 98.0894 |
| V8 | 0.9393 | 0.0682 | 0.1668 | 93.5948 |

**6 Rounds of Training**

|  | R2 | MSE | MAE | Time/s |
|---|---|---|---|---|
| V1 | 0.9288 | 0.0684 | 0.1720 | 136.4278 |
| V2 | 0.9295 | 0.0669 | 0.1685 | 145.1813 |
| V3 | 0.9351 | 0.0667 | 0.1629 | 61.5993 |
| V4 | 0.9333 | 0.0677 | 0.1655 | 64.7559 |
| V5 | 0.9355 | 0.0651 | 0.1614 | 88.2861 |
| V6 | 0.9448 | 0.0661 | 0.1627 | 95.2478 |
| V7 | 0.9377 | 0.0664 | 0.1628 | 96.6539 |
| V8 | 0.9473 | 0.0677 | 0.1621 | 88.7919 |

**Table 6. Prediction indicators for unit 2 of Longyuan wind farm.**

**3 Rounds of Training**

|    | R2 | MSE | MAE | Time/s |
|----|-----|-----|-----|--------|
| V1 | 0.9173 | 0.0788 | 0.1951 | 107.2609 |
| V2 | 0.9235 | 0.0730 | 0.1818 | 118.9235 |
| V3 | 0.9182 | 0.0812 | 0.1962 | 60.9445 |
| V4 | 0.9160 | 0.0820 | 0.2000 | 59.4605 |
| V5 | 0.9263 | 0.0770 | 0.1847 | 86.5272 |
| V6 | 0.9203 | 0.0749 | 0.1886 | 86.9764 |
| V7 | 0.9198 | 0.0813 | 0.1969 | 86.7902 |
| V8 | 0.9284 | 0.0738 | 0.1874 | 85.4334 |

**6 Rounds of Training**

|    | R2 | MSE | MAE | Time/s |
|----|-----|-----|-----|--------|
| V1 | 0.9238 | 0.0717 | 0.1820 | 120.1294 |
| V2 | 0.9288 | 0.0736 | 0.1814 | 120.9358 |
| V3 | 0.9159 | 0.0764 | 0.1913 | 58.4095 |
| V4 | 0.9243 | 0.0755 | 0.1870 | 60.8611 |
| V5 | 0.9199 | 0.0748 | 0.1865 | 86.4150 |
| V6 | 0.9353 | 0.0729 | 0.1814 | 89.5969 |
| V7 | 0.9237 | 0.0723 | 0.1813 | 87.1209 |
| V8 | 0.9315 | 0.0725 | 0.1838 | 88.4992 |

**Table 7. Prediction indicators for unit 13 of Longyuan Wind Farm.**

**3 Rounds of Training**

|    | R2 | MSE | MAE | Time/s |
|----|-----|-----|-----|--------|
| V1 | 0.8881 | 0.0863 | 0.2140 | 123.5782 |
| V2 | 0.9196 | 0.0731 | 0.1854 | 123.6062 |
| V3 | 0.9295 | 0.0696 | 0.1739 | 60.7259 |
| V4 | 0.9307 | 0.0765 | 0.1877 | 59.8004 |
| V5 | 0.9309 | 0.0687 | 0.1761 | 81.3101 |
| V6 | 0.9258 | 0.0728 | 0.1803 | 85.2531 |
| V7 | 0.9298 | 0.0705 | 0.1783 | 85.4746 |
| V8 | 0.9391 | 0.0706 | 0.1797 | 85.4554 |

**6 Rounds of Training**

|    | R2 | MSE | MAE | Time/s |
|----|-----|-----|-----|--------|
| V1 | 0.9334 | 0.0665 | 0.1681 | 136.8194 |
| V2 | 0.9090 | 0.0801 | 0.2017 | 118.8172 |
| V3 | 0.9349 | 0.0661 | 01663 | 90.8462 |
| V4 | 0.9295 | 0.0686 | 0.1705 | 69.1581 |
| V5 | 0.9301 | 0.0675 | 0.1722 | 99.4972 |
| V6 | 0.9310 | 0.0672 | 0.1684 | 98.7927 |
| V7 | 0.9317 | 0.0661 | 0.1680 | 94.2851 |
| V8 | 0.9370 | 0.0707 | 0.1743 | 97.6012 |

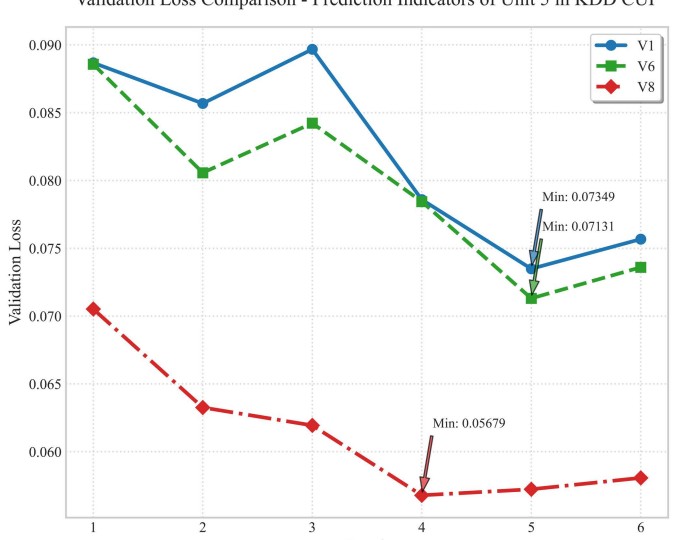
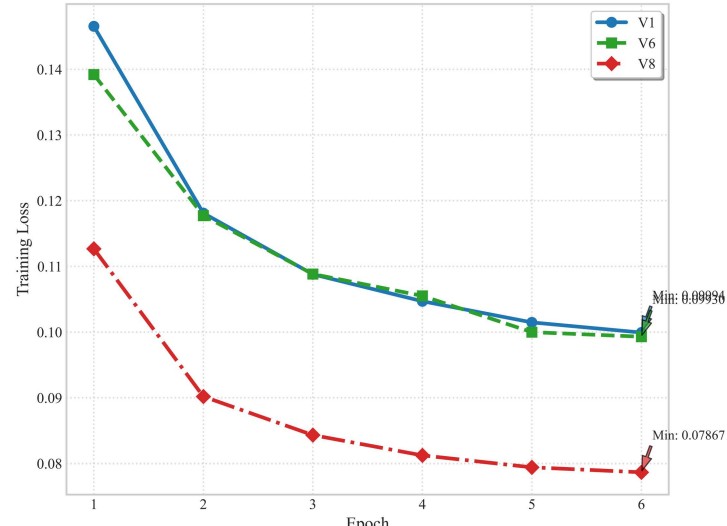

**Fig 15. Prediction Results of 6 Rounds of Training for Unit 13 of Longyuan Power Plant.**

### 4.3. Comparison of model convergence and optimization effect

The Fig 15 are derived from the prediction indicator data of Unit 5 in the KDD CUP dataset, presenting the variation trends of validation loss and training loss for three models (V1, V6, V8) across 6 training epochs (Epoch). In terms of validation loss (left chart), the V1 model shows significant fluctuations, with values ranging from approximately 0.07 to 0.09, while the V8 model exhibits a relatively stable and continuously decreasing trend, with the minimum value reaching 0.05679, far lower than V1's minimum of 0.07131. For training loss (right chart), the V8 model also demonstrates a more rapid and steady decline, with the final loss approaching 0.08, notably outperforming V1 and V6. Evidently, the optimized V8 model, compared to the baseline V1 and partially optimized V6, exhibits superior convergence speed and loss control, with the loss reduction effect being particularly prominent, fully verifying the effectiveness of the MFIO – Informer optimization strategy in wind power prediction for Unit 5 of the KDD CUP dataset.

## 5. Conclusion

By leveraging multi-dimensional data for wind turbine health assessment, a multi-dimensional interactive health matrix is derived. Optimization of the encoder, decoder, embedding vectors, and prediction process in Informer accelerates model convergence while enhancing prediction accuracy and speed. Simulation tests demonstrate that the encoder integrated with the optimized Informer model exhibits superior prediction performance. Compared to the traditional Informer, the proposed MFIO-Informer achieves better optimization effects in wind power prediction with lower data costs.

## Author contributions

**Conceptualization:** Feng Huang, Bing Wei.

**Data curation:** 周文娟 周.

**Formal analysis:** 周文娟 周.

**Funding acquisition:** Bing Wei.

**Investigation:** Feng Huang, Liang Li.

**Methodology:** Liang Li.

**Project administration:** Shixi Dai, Xin Xie.

**Resources:** Shixi Dai.

**Software:** Xin Xie, Youyuan Peng.

**Supervision:** Xin Xie.

**Visualization:** Hong You.

**Writing – original draft:** 周文娟 周.

**Writing – review & editing:** 周文娟 周.

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
