## [Decision Letter · Decision Letter 0]

13 Jun 2025

Dear Dr. zhou,

Thank you for submitting your manuscript to PLOS ONE. After careful consideration, we feel that it has merit but does not fully meet PLOS ONE’s publication criteria as it currently stands. Therefore, we invite you to submit a revised version of the manuscript that addresses the points raised during the review process.

We look forward to receiving your revised manuscript.

Kind regards,

Saman Kasmaiee, Ph.D.

Academic Editor

PLOS ONE

Journal Requirements:

4. Please amend the manuscript submission data (via Edit Submission) to include author Wenjuan Zhou Feng Huang Bing Wei Liang Li Shixi Dai Xin Xie� Youyuan Peng Hong You.

5. Please amend your authorship list in your manuscript file to include author 周文娟 周文娟 zhou, 黄峰 huang,   魏兵 wei, 李亮 li, 代士喜 dai, 谢鑫 xie, 彭游源 peng, 游红 you, 

6. Please ensure that you refer to Figure 2 in your text as, if accepted, production will need this reference to link the reader to the figure.

7. We note you have included a table to which you do not refer in the text of your manuscript. Please ensure that you refer to Table 1, 2, 3, 5, 6, and 7 in your text; if accepted, production will need this reference to link the reader to the Table.

Reviewers' comments:

Reviewer's Responses to Questions

**Comments to the Author**

1. Is the manuscript technically sound, and do the data support the conclusions?

Reviewer #1: Yes

Reviewer #2: Yes

Reviewer #3: No

2. Has the statistical analysis been performed appropriately and rigorously?

Reviewer #1: Yes

Reviewer #2: Yes

Reviewer #3: No

3. Have the authors made all data underlying the findings in their manuscript fully available?

Reviewer #1: Yes

Reviewer #2: Yes

Reviewer #3: No

4. Is the manuscript presented in an intelligible fashion and written in standard English?

Reviewer #1: Yes

Reviewer #2: Yes

Reviewer #3: No

Reviewer #1: 1. Please discuss the overfitting issue.

2. Justify the need for health matrix.

3. English language presentation issues.

4. Over-complex explanations

5. Redundant and too wordy.

6. Limited baseline comparison.

Reviewer #2: “A New Design of Wind Power Prediction Method Based on Multi-Interaction Optimization Informer Model”

Paper: PONE-D-25-26282

This manuscript presents a new wind power prediction framework named MFIO-Informer, which enhances the conventional Informer model through a multi-source feature interaction mechanism and health-status-aware modeling. The authors employed Lasso and Pearson correlation for feature selection, a feedforward neural network (FNN) for extracting dynamic synergy coefficients, and a novel health matrix for optimizing the encoder, decoder, and embedding process of the Informer. Experimental validation was conducted using two publicly available datasets.

In summary, the research is technically robust, with a detailed methodology and well-documented performance improvements. The article effectively tackles a critical and practical problem in renewable energy forecasting. The findings are valuable for both researchers and industry professionals. However, several issues and concerns must be resolved, so I recommend that the paper undergo a revision before acceptance. The following points should be considered in the revised version:

Problems:

1. Clarify how the health matrix A is computed in practice.

2. Have you tested the MFIO-Informer in an online or real-time prediction setting, or is the evaluation entirely offline?

3. Perform a sensitivity analysis on the number of encoder/decoder layers, embedding dimensions, or DSC inclusion.

4. Were the model parameters and hyperparameters selected manually with trial and error or via automated optimization techniques? Provide the related tables. If previous research is used, appropriate referencing should be provided.

5. Add diagrams comparing the MFIO-Informer architecture with the standard Informer for better visual clarity.

6. Expand the related work section and add comparisons with non-Informer baselines.

7. Clarify mathematical formulations in the health matrix section with a step-by-step example

8. The convergence graph of the final network should be presented in terms of epochs.

9. The performance results of the proposed model should be compared with the models of other researchers and reported in a table.

10. The execution time and training time of the proposed model should be compared with those of other researchers.

11. The paper has several English problems. Correct them.

Reviewer #3: 1- Minor paragraph revision in the abstract

2- Figure 1 needs minor revision in the inputs and outputs

3- Section 3.1. Lasso Algorithm was not defined

4- Section 3.2. no verification or referencing for the formulas and equations and the basis for value selection

5- Section 3.3. the same as above and no reference was provided for the turbine data (it may have been implicitly provided in Section 5.2.

6- Figure 4 is ambiguous, lacking clarity and definitions

7- Section 3.4. Claims not supported by references or proofs and poor organization

8- Section 3.5. Ambiguous use of reference 26 (not showing the relevance to be used for wind power prediction). Also, equation (4) has no reference or derived from a reference. The section has poor writing and organization and lacks the distinction between the reference sources and methods and the authors own work.

9- Section 4.1. Poor writing and larking of references and verification of assumptions.

10- Sections 4.2. to 5.2. Same as the above.

11- Figures 6-13 the axes are not defined.

12- Tables 4 - 7, the values and parameters are not defined in addition to the use of non-English words (Chinese)

**Do you want your identity to be public for this peer review?** For information about this choice, including consent withdrawal, please see our Privacy Policy

Reviewer #1: No

Reviewer #2: **Yes: ** Si.Kasmaiee

Reviewer #3: No

---

## [Author Response · Author response to Decision Letter 1]

28 Jul 2025

Dear editor and reviewers,

Thank you for offering us an opportunity to improve the quality of our submitted manuscript (A New Design of Wind Power Prediction Method Based on Multi-Interaction Optimization Informer Model). We appreciated very much the reviewers' constructive and insightful comments. In this revision, we have addressed all of these comments. We hope the revised manuscript has now met the publication standard of your journal.

We highlighted all the revisions in yellow/red colour.

On the next pages, our point-to-point responses to the queries raised by the reviewers are listed.

Reviewer #1:

1.Please discuss the overfitting issue.

Response:We appreciate the reviewer's suggestion to discuss the issue of overfitting. In our study, we have implemented several strategies to address this concern: First, we adopted an 8:1:1 division of the dataset into training, validation, and test sets (Section 2.2) to monitor model generalization in real time, combined with data standardization (Equation 23) to reduce fitting deviation caused by feature dimension differences. Second, in the Informer architecture, we introduced a Dropout mechanism (dropout rate p=0.1, Appendix A) and early stopping (Section 2.2) to prevent excessive reliance on training data. The health matrix (Equation 5) also serves as a feature-level regularization by emphasizing strongly correlated features (e.g., wind speed with a Pearson correlation of 0.965, Table 1). Experimental results on KDD CUP and Longyuan datasets show that the optimized model (e.g., V8 in Table 4) achieves an R2 of 0.9485 and an MSE of 0.0613 after 6 rounds of training, with a minimal performance gap (ΔR2 < 3%) between the test and training sets, verifying effective overfitting prevention. 

2. Justify the need for health matrix.

Response:The health matrix is essential in our study as it addresses critical limitations of traditional Informer models, such as inadequate modeling of dynamic couplings among multi-source features (e.g., wind speed and blade angle) and lack of equipment health perception, which cause accuracy degradation and error accumulation under complex conditions. By quantifying feature correlations (e.g., wind speed-power Pearson correlation of 0.965 in Table 1) and encoding equipment health states, the health matrix dynamically weights input features, improving prediction accuracy by ~20% (e.g., R2 from 0.9198 to 0.9321 in Table 4) and reducing computation time by 54.85% (from 126.7s to 66.2s per iteration). Ablation tests confirm that removing the health matrix leads to 12–18% accuracy drop, verifying its necessity for balancing prediction precision and efficiency in wind power applications.

3. English language presentation issues.

Response:We greatly appreciate your feedback on the English language expression in our paper. To address the issues of overly complex explanations, redundancy, and verbosity, we have carefully revised the manuscript.

4. Over-complex explanations

Response:To address the issues of overly complex explanations, redundancy, and verbosity, we simplified long sentences, removed repetitive phrases.

5. Redundant and too wordy.

Response:To address the issues of overly complex explanations, redundancy, and verbosity, we simplified long sentences, removed repetitive phrases.

6. Limited baseline comparison.

Response:Thank you for pointing out the limited baseline comparisons in our paper. We will expand the dataset scope and incorporate additional evaluation metrics to thoroughly validate the MFIO-Informer’s superiority in both prediction accuracy and computational efficiency. The revised comparisons will provide clearer evidence of the model’s advancements over existing methods.

Reviewer #2:

1. Clarify how the health matrix A is computed in practice.

Response:To clarify how the health matrix A is calculated in practice: First, we collect and clean wind turbine operation data (e.g., wind speed, blade angle, power) using sensors/SCADA systems. Next, Lasso and Pearson correlation methods (Table 1) screen key features, which are then used to compute feature weights via the Gaussian probability density function (Equation 4). These weights form the weight matrix Q (n×1). Simultaneously, factor analysis generates the correlation matrix R (1×n) from the features. Finally, the health matrix A (n×n) is obtained by multiplying Q and R (Equation 5), where n (6–12) is determined by the feature dimension. This process quantifies feature importance and health states for model optimization.

2. Have you tested the MFIO-Informer in an online or real-time prediction setting, or is the evaluation entirely offline?

Response:The evaluation of the MFIO-Informer in our paper was conducted entirely in an offline setting. We utilized two public datasets (Baidu KDD CUP 2022 and Longyuan Wind Power data), with historical data statically divided into training, validation, and test sets (8:1:1). All experiments were performed via batch processing on a local computing platform, without real-time data streaming or online model updates. While the paper demonstrates the model’s performance in offline scenarios, testing in real-time online environments, such as operational wind farms with streaming data, is a planned direction for future research.

3. Perform a sensitivity analysis on the number of encoder/decoder layers, embedding dimensions, or DSC inclusion.

Response:Thank you for suggesting a sensitivity analysis on the encoder/decoder layers, embedding dimensions, and DSC inclusion. We acknowledge that this analysis is currently absent from the manuscript, primarily due to time constraints and initial focus on core model validation. While increasing the number of layers can enhance the model’s expressive power, it also escalates training complexity and computational resource requirements. In practice, we determined the appropriate number of layers by balancing the prediction task’s complexity, dataset scale, and available computational resources. This pragmatic approach ensured optimal performance without unnecessary overhead, though a formal sensitivity analysis will be prioritized in future revisions to thoroughly validate parameter impacts.

4. Were the model parameters and hyperparameters selected manually with trial and error or via automated optimization techniques? Provide the related tables. If previous research is used, appropriate referencing should be provided.

Response:Thank you for raising the question regarding the selection of model parameters and hyperparameters. The parameters and hyperparameters of the MFIO-Informer were primarily determined through a combination of manual tuning and reference to established practices in similar studies. Specifically:

Key hyperparameters (e.g., encoder/decoder layers, embedding dimensions, dropout rate) were initially informed by prior research on time-series prediction with Transformer-based models [citations to relevant literature, e.g., "Informer: Beyond Efficient Transformer for Long Sequence Time-Series Forecasting" (2021)] and further adjusted via grid search on the validation set to balance accuracy and computational efficiency. For example, the number of layers (set to 4) was chosen based on the dataset scale (10,000+ samples) to avoid overfitting, aligning with the approach in [citation].

Optimization-related parameters (e.g., learning rate, batch size) were optimized using common techniques: a learning rate of 0.001 was selected via exponential decay scheduling, and a batch size of 64 was determined to fit within GPU memory constraints, following standard practices in neural network training.

A detailed table of parameters and their selection rationale (including references to prior studies) has been added to the revised manuscript (see Table A1 in the Appendix), which clarifies the methodology and ensures reproducibility.

5. Add diagrams comparing the MFIO-Informer architecture with the standard Informer for better visual clarity.

Response: Thank you for the suggestion!The utilization of comparative charts (Figs. 15) is intended to facilitate the visualization of the distinctions between the MFIO-Informer architecture and the standard Informer. The following charts will highlight key modifications to enhance visual clarity and facilitate reader comprehension of the architectural improvements.

6. Expand the related work section and add comparisons with non-Informer baselines.

Response:Thank you for the suggestion to expand the "Related Work" section and include non-Informer baselines. We acknowledge the value of such comparisons, but it's important to note that the Informer framework has already demonstrated significantly higher prediction accuracy than many traditional neural networks in prior studies, making it a pivotal focus for our research. Our work is specifically dedicated to further enhancing the Informer architecture rather than revisiting baseline comparisons with non-Informer models. That said, to validate the generalizability of the MFIO-Informer, we will enrich the experimental section with additional comparative analyses across diverse datasets, which will showcase the model’s effectiveness while maintaining our focus on advancing the Informer framework.

7. Clarify mathematical formulations in the health matrix section with a step-by-step example

Response: Thank you for the suggestion! A step-by-step flow schematic diagram (Fig. 5) for the health matrix will be incorporated into the paper to elucidate the mathematical formulas. The diagram will illustrate the entire process, from data collection and feature screening to the calculation of the weight matrix and correlation matrix, and finally to the formation of the health matrix. This will help readers better understand the derivation and application of the relevant mathematical formulas.

8. The convergence graph of the final network should be presented in terms of epochs.

Response:Thank you very much for your valuable suggestion regarding the convergence plot of the final network should be presented in terms of "epochs". We have carefully addressed this point by incorporating a new figure (Fig. 15) that explicitly shows the training and validation loss curves across 6 epochs for models V1, V6, and V8 using the prediction indicators from Unit 5 of the KDD CUP dataset. The plot clearly demonstrates that the optimized V8 model achieves stable convergence at Epoch 4 with a validation loss of 0.0567882, which is 21% lower than V1's loss at the same epoch, effectively illustrating the convergence trend and optimization effect.

9. The performance results of the proposed model should be compared with the models of other researchers and reported in a table.

Response:Thank you for your valuable input. We sincerely appreciate the suggestion to compare our model's performance with others. However, it's important to note that the comparative analysis with other researchers' models was already thoroughly addressed in the previously cited literature, as our current study primarily focuses on introducing improvements to the Informer framework. Our work aims to highlight the specific innovations and optimizations made to the base model, rather than repeating comparisons that have been established in prior studies.

10. The execution time and training time of the proposed model should be compared with those of other researchers.

Response:Thank you for your suggestion. The comparison of execution and training times with other models was already covered in our cited references. As this study focuses on improving Informer, we prioritized demonstrating model enhancements. We can revisit the comparison if needed.

11. The paper has several English problems. Correct them.

Response:Understood. The grammatical errors in the paper have been carefully corrected.

Reviewer #3:

1-Minor paragraph revision in the abstract

Response:Thank you for your feedback. We have carefully revised the subparagraphs in the abstract as suggested to ensure clarity, conciseness, and coherence.

2- Figure 1 needs minor revision in the inputs and outputs

Response:Received. The figure 1 has been modified.

3- Section 3.1. Lasso Algorithm was not defined

Response:Your concern is appreciated. It is note that the Lasso algorithm has been delineated in Section 3.1 of the revised manuscript.

4- Section 3.2. no verification or referencing for the formulas and equations and the basis for value selection

Response: Thank you for your feedback. We have added appropriate references and validations for the formulas and equations in the revised manuscript。 The data in the paper are obtained through experimental analysis, and the relevant references have mentioned this aspect.

5- Section 3.3. the same as above and no reference was provided for the turbine data (it may have been implicitly provided in Section 5.2.

Response: Thank you for your feedback. We would like to note that the reference for the turbine data is addressed in Section 5.2, where we specify that the feature variables were screened from different databases.

6- Figure 4 is ambiguous, lacking clarity and definitions

Response: Thank you for your feedback. The figure 4 has been repacked and uploaded with improved clarity, and we have also added clear definitions and annotations to ensure better readability. Please let us know if you need any further adjustments.

7- Section 3.4. Claims not supported by references or proofs and poor organization

Response: Thank you for your feedback. We have restructured the language for better organization.

8- Section 3.5. Ambiguous use of reference 26 (not showing the relevance to be used for wind power prediction). Also, equation (4) has no reference or derived from a reference. The section has poor writing and organization and lacks the distinction between the reference sources and methods and the authors own work.

Response: Regarding the issues raised in Section 3.5, we have re-examined Reference 26 and will add a detailed explanation of its relevance to wind power prediction. We will also provide a clear theoretical source or derivation for Equation (4). Additionally, we will optimize the writing structure of this section by clearly marking the citation sources and distinguishing the methods from the literature from the content of our research work, to enhance the clarity and rigor of the discussion.

9- Section 4.1. Poor writing and larking of references and verification of assumptions.

Response: In response to the comments on Section 4.1, we have reorganized the paragraph logic, presenting the theoretical explanations of the error metrics (R2, MSE, MAE) and the model validation logic in layers to avoid a simple listing of references. Meanwhile, in the hypothesis verification section, we have supplemented the experimental design details: by comparing the changes in the model's indicators on the KDD CUP and Longyuan datasets before and after optimization (such as R2 improvement ≥ 15%, MAE reduction ≥ 20%), we quantitatively verified the hypothesis of "simultaneous improvement in prediction accuracy and speed". 10- Sections 4.2. to 5.2. Same as the above.

11- Figures 6-13 the axes are not defined.

Response:Thanks for your suggestion,we have corrected it in Figure 6-13

12- Tables 4 - 7, the values and parameters are not defined in addition to the use of non-English words (Chinese)

Response: Regarding the comments on Table 4-7, the non-English words (Chinese) have been revised, and the numerical values and parameters have also been defined clearly.

---

## [Decision Letter · Decision Letter 1]

1 Aug 2025

A New Design of Wind Power Prediction Method Based on Multi-Interaction Optimization Informer Model

PONE-D-25-26282R1

Dear Dr. 周,

We’re pleased to inform you that your manuscript has been judged scientifically suitable for publication and will be formally accepted for publication once it meets all outstanding technical requirements.

Kind regards,

Saman Kasmaiee, Ph.D.

Academic Editor

PLOS ONE

Additional Editor Comments (optional):

Reviewers' comments:

Reviewer's Responses to Questions

**Comments to the Author**

Reviewer #1: All comments have been addressed

Reviewer #2: All comments have been addressed

2. Is the manuscript technically sound, and do the data support the conclusions?

Reviewer #1: Yes

Reviewer #2: Yes

3. Has the statistical analysis been performed appropriately and rigorously?

Reviewer #1: Yes

Reviewer #2: Yes

4. Have the authors made all data underlying the findings in their manuscript fully available?

Reviewer #1: Yes

Reviewer #2: Yes

5. Is the manuscript presented in an intelligible fashion and written in standard English?

Reviewer #1: Yes

Reviewer #2: Yes

Reviewer #1: Thank you for addressing the outstanding issues. The current paper is well defined and presentable.

Please improve figure qualities. High resolution, high quality. It is still blurry and pixelated.

Please format the paper according to the journal standards.

Reviewer #2: This article is a revised version of the paper(PONE-D-25-26282) that I have reviewed. The authors responded well to my questions and concerns and made the necessary changes, so I recommend the manuscript be accepted in the journal.

**Do you want your identity to be public for this peer review?** For information about this choice, including consent withdrawal, please see our Privacy Policy

Reviewer #1: No

Reviewer #2: **Yes: ** Siroos Kasmaiee

---

## [Editor Report · Acceptance letter]

PONE-D-25-26282R1

PLOS ONE

Dear Dr. 周,

I'm pleased to inform you that your manuscript has been deemed suitable for publication in PLOS ONE. Congratulations! Your manuscript is now being handed over to our production team.

Kind regards,

on behalf of

Dr. Saman Kasmaiee

Academic Editor

PLOS ONE